# A Role for Caveolin-3 in the Pathogenesis of Muscular Dystrophies

**DOI:** 10.3390/ijms21228736

**Published:** 2020-11-19

**Authors:** Bhola Shankar Pradhan, Tomasz J. Prószyński

**Affiliations:** Łukasiewicz Research Network–PORT Polish Center for Technology Development, 147 Stabłowicka Street, 54-066 Wrocław, Poland; Bhola.Pradhan@port.lukasiewicz.gov.pl

**Keywords:** caveolae, Caveolin-3, scaffolding domain, endocytic pathway, muscular dystrophy, cardiovascular diseases, dystrophin, dystroglycan

## Abstract

Caveolae are the cholesterol-rich small invaginations of the plasma membrane present in many cell types including adipocytes, endothelial cells, epithelial cells, fibroblasts, smooth muscles, skeletal muscles and cardiac muscles. They serve as specialized platforms for many signaling molecules and regulate important cellular processes like energy metabolism, lipid metabolism, mitochondria homeostasis, and mechano-transduction. Caveolae can be internalized together with associated cargo. The caveolae-dependent endocytic pathway plays a role in the withdrawal of many plasma membrane components that can be sent for degradation or recycled back to the cell surface. Caveolae are formed by oligomerization of caveolin proteins. Caveolin-3 is a muscle-specific isoform, whose malfunction is associated with several diseases including diabetes, cancer, atherosclerosis, and cardiovascular diseases. Mutations in Caveolin-3 are known to cause muscular dystrophies that are collectively called caveolinopathies. Altered expression of Caveolin-3 is also observed in Duchenne’s muscular dystrophy, which is likely a part of the pathological process leading to muscle weakness. This review summarizes the major functions of Caveolin-3 in skeletal muscles and discusses its involvement in the pathology of muscular dystrophies.

## 1. Introduction

The caveolin family of proteins is comprised of three members: Caveolin-1 (Cav1), Caveolin-2 (Cav2), and Caveolin-3 (Cav3). Cav1 was the first caveolin discovered as a substrate of tyrosine kinase Src [1]. Cav2 was identified in adipocyte membrane fraction [2], and Cav3, which is specific to muscle cells, was identified based on sequence similarity to Cav1 [3,4]. Cav1 and 2 form hetero-oligomers that can aggregate into organized structures at the invaginated membrane pits called caveolae, which in Latin means “little caves” [5,6]. Caveolae are present in many cell types including adipocytes, endothelial cells, fibroblasts, smooth-muscles, skeletal and cardiac muscles, and epithelial cells. Caveolae have multiple functions that involve the organization of specific microdomains at the plasma membrane, the regulation of signaling, endocytosis, exocytosis, and mechano-transduction [7,8,9,10]. Mutations in caveolin proteins and malfunction of caveolae are associated with several diseases [11,12,13,14,15,16]. Mutations in muscle-specific isoform Cav3 lead to muscular dystrophies characterized by muscle weakness, an increased number of centralized nuclei, and a high level of serum creatine kinase [13,17,18]. Caveolin-3 is known to associate with several components of the multiprotein dystrophin glycoprotein complex (DGC). Malfunction of most of the DGC leads to muscular dystrophies, such as Duchene muscular dystrophy (DMD), caused by a lack of functional dystrophin [19,20]. DMD affects one in 3500 newborn boys worldwide and the main characteristic features are muscle dystrophy, respiratory insufficiency, cardiomyopathy, and premature death [21]. The lack of functional dystrophin in DMD is due to the deletions, duplications, or point mutations in the open reading frame of the dystrophin gene. There are obvious differences in the pathology of mice models of muscular dystrophy and that in human patients. One reason for this could be that mice have higher expression of utrophin, which could partially compensate for the lack of functional dystrophin [22]. On the other hand, there are also variations in the pathology of muscular dystrophies in human patients. Elevated Cav3 expression has been observed in DMD patients and a mouse model of this disorder (mdx mice) [19]. This suggests that Cav3 could play an important role in the pathogenesis of skeletal muscles and could be a potential pharmaceutical target. Further research should be conducted in this direction.

## 2. Domain Organization and Structure of Caveolins

The three caveolins have similar domain composition. The C’- and N’-terminal portion of the proteins are exposed to the cytoplasm and the middle hydrophobic part makes a single loop which is inserted into the lipid bilayer without crossing through it (Figure 1) [23,24]. The cytoplasmic C’-terminal region has an additional three palmitoylated sites. The juxta membrane part of the N’-terminal cytoplasmic tail contains the caveolin scaffolding domain (CSD), which interacts with many signaling proteins. This region is located within the oligomerization motif, which aggregates caveolins into larger assemblies (Figure 1). Nine Cav3 monomers assemble to form a toroidal shape complex of ~16.5 nm in diameter and ~5.5 nm in height [25]. The nonamers of caveolins assemble into larger polyhedral-like structures, triggering membrane invagination into pits called caveolae. The process of membrane invagination is somehow similar to that regulated by clathrin. Caveolins can, therefore, be present at the membranes in three different states: monomers, nonamers, and larger clusters of nonamers which could have either flat shapes or are curved into caveolae with invaginated membranes.

Within the membrane inserted region, caveolins have a highly hydrophobic sequence which binds cholesterol. Oligomerization of caveolins and interactions with cholesterol create a unique environment in the membrane which triggers accumulation of sphingolipids at the outer leaflet of the bilayer and formation of a specialized lipid raft [26,27,28,29,30]. Lipid rafts have been proposed to form microdomains on the cell surface which recruit a specific set of proteins and lipids, while other components of the plasma membrane are excluded based on physical properties [31]. These platforms at the cell surface facilitate protein interactions and signaling. Caveolin-rich domains, apart from sphingolipids and cholesterol, are also enriched in phosphatidylinositol (4,5)-bisphosphate (PtdIn(4,5)P2), which regulates many signaling proteins [32]. Caveolins at the cell surface can be scaffolded by actin and microtubule cytoskeletons regulating their stability [9,33,34].

Formation of caveolae is controlled by a family of caveolin-interacting proteins called cavins. There are four cavin proteins identified. Cavin1 targets other cavin proteins to the membrane [35], while Cavin2 mediates curving of the membrane into caveolae or tubules [36]. Cavin3 mediates endocytosis of caveolae and their intracellular transport [37]. Cavin4 is expressed predominantly in muscle cells. This cavin isoform appears to be dispensable for caveolae formation but could be involved in the recruitment of caveolae-associated signaling proteins like the extracellular signal-regulated kinase 1 and 2 (ERK1/2) [7].

## 3. Function of Caveolins in Regulation of Endocytosis and Exocytosis

Internalization of caveolae and endocytosis of associated cargo involves Dynamin GTPase [38], similar to endocytosis of clathrin-coated vesicles. Upon internalization, the endocytosed vesicles on the caveolin-dependent pathway can fuse with endosomes from which the associated cargo could be degraded in the lysosomal pathway or sent back to the plasma membrane through the recycling pathway. Interestingly, internalization of the cell surface receptors does not necessarily block their signaling activity. Recent findings suggest that much receptor-dependent signaling occurs also in the internalized endocytic vesicles [39]. For instance, the epidermal growth factor receptors (EGFR) interact with growth factor receptor-bound protein 2 (Grb2), which initiates Ras and mitogen-activated protein kinases (MAPK) signaling at the endosomes from where the signal transduction is initiated [40]. The caveolin-dependent endocytic pathway provides an entry route for several pathogenic viruses. For instance, a widely studied Simian virus 40 (SV40) enters cells through caveolae-mediated endocytosis [41]. A similar mechanism for cell infection has been reported recently for Zika virus causing microcephaly and neurological disorders [42]. However, the rate at which caveolae bud from the plasma membrane has been proposed to be very low (at least at steady state), raising questions about the overall contribution of the caveolae mediated endocytosis to total endocytosis happening inside the cell [43,44]. Further studies are needed to completely understand the cellular functions of caveolae mediated endocytosis.

Caveolins can also be involved in autophagy. Cav3 has been shown to play a critical role in autophagy, protecting the cardiac muscle from damage after ischemia and reperfusion [45,46]. Interestingly, Cav3 interacts with Beclin-1 and Atg12, the important proteins involved in autophagy [45].

Caveolins could also be implicated in the regulation of exocytosis. In the exocytic pathway, caveolins are clustered into specific secretory vesicles at the trans-Golgi network where protein sorting takes place. During vesicle formation, caveolins could capture associated proteins and lipids into the forming transport containers. For instance, exocytosis of the angiotensin II type 1 receptor has been shown to be regulated by Caveolin-1 [47].

## 4. Function of Caveolins in Regulation of Signaling

Apart from oligomerization, caveolin proteins can directly interact with a large number of signaling proteins including EGFR, platelet-derived growth factor (PDGF) receptors, insulin receptors, Shc, Grb-2, mSOS-1, nitric oxide synthases (NOS), H-Ras, Raf, MEK, and ERK1/2 [8]. Most of these interactions are mediated by the caveolin scaffolding domain (Figure 1) and occur in caveolae. Caveolins sequester many binding-partners in their inactive states, as is the case for H-Ras, Gα subunits, and Src [48]. Moreover, activation of Gα (and other proteins) has been shown to abolish the interactions with caveolins and trigger re-localization of activated forms away from caveolae. As a consequence, caveolae serve as reservoirs for storage of inactivated signaling molecules or pockets at the plasma membrane where inactivated cargo is collected for subsequent endocytosis. The opposite situation exists with some G-protein coupled receptors, like B2 bradykinin receptors or the muscarinic acetylcholine receptor [49]. They translocate to caveolae when activated by ligands or agonists. For instance, the muscarinic acetylcholine receptor in cardiomyocytes translocate to Cav3-containing caveolae when activated by its agonist carbachol [49].

In skeletal muscles, the PI3K-Akt-mTOR pathway plays an important role in the regulation of muscle growth and hypertrophy [50,51,52]. This is particularly important in muscle wastage after injury, during aging, in muscular dystrophies, and cancer-mediated cachexia [52,53,54]. There have been reports suggesting that Akt signaling could be regulated by caveolins. In HEK-293 and HeLa cells, Cav1 overexpression leads to increased Akt activity, possibly through the diminished synthesis of ceramides which inhibit PI3K [55,56]. Moreover, in human SV589 fibroblasts and mouse embryonic fibroblasts, the loss of Cavin3 promoted Akt signaling and Cavin3 overexpression inhibited Akt signaling [57].

Caveolins’ interaction with many signalling proteins was shown to be mediated by the CSD. There are also studies that call these interactions into question, arguing that the CSD is too close to membrane that may block their interactions with cytoplasmic proteins [58,59].

## 5. Caveolae as Mechanosensors

Mechanosensing refers to mechanisms that cells use to detect and respond to internal and external forces. In 1975, Dulhunty et al. [60] proposed that muscle caveolae would breakopen in response to stretching of skeletal muscle fibers. In this process, caveolae serve as a reservoir of membrane located in the invaginations and internal vesicles which can be quickly released in response to tension on the plasma membrane. Remarkably, in muscle cells, mechanoprotection by caveolae is directly coupled with IL6/STAT3 signaling, which malfunctions in Cav3-associated dystrophies [10]. Caveolin-mediated mechanosensing is not limited to muscle cells and has been reported in many cell types in vitro and in vivo [61,62,63]. It has been shown that caveolae present in the vascular endothelial cell act as mechanosensors of the blood flow and respond to increased blood pressure by disassembly of caveolae, release of accumulated membrane, and decrease in tension [64,65,66,67].

## 6. Function of Caveolin-3 in Muscle Development and Repair

Caveolin-3 regulates several processes that play important roles at various stages of skeletal muscle development and repair [68,69,70]. Its expression is required for differentiation and efficient fusion of myoblast and myotube formation [69,70,71]. The targeted downregulation of Cav3 in C2C12 myoblasts by expressing Cav3 antisense RNA is sufficient to inhibit myoblast fusion [72]. Madaro et al. [73] have reported that PKCθ signaling is required for myoblast fusion by regulating the expression of Cav3 and β1D integrin.

Corrotte et al. [74] have shown that cellular damage triggers an influx of calcium ions, which leads to the exocytosis of lysosomal compartments and the release of acidic sphingomyelinase. This enzyme converts sphingomyelin to ceramides and ceramides-enriched microdomains promote invagination of the lipid bilayer. Caveolae concentrate at the site of injury and the resorption of material is mediated through caveolae-dependent endocytosis. The processes of caveolae-mediated repair are inhibited in caveolinopathies when mutations in caveolins compromise their functions. Caveolin-3 interacts with dysferlin (DYSF) and TRIM 72, two proteins involved in muscle repair. Dysferlin likely regulates Ca2+-dependent fusion of the repair exotic vesicle with the plasma membrane at the sites of muscle fibre injury [75]. Defects in the DYSF gene result in several types of muscular dystrophy; including limb-girdle muscular dystrophy type 2B (LGMD2B), Distal Myopathy (DM), and Miyoshi myopathy (MM). Interaction with Cav3 stabilises dysferlin at the cell surface [76]. In the absence of Cav3, dysferlin is endocytosed at a rapid rate and membrane repair is compromised. TRIM72, a muscle-specific TRIM (Tri-partite motif) family protein, contributes to intracellular vesicle trafficking and is an essential component of the membrane repair machinery in striated muscles [77]. Cav3 mutations resulting in the retention of Cav3 at the Golgi lead to mislocalization of TRIM72 and dysferlin and to defects in membrane repair [68].

When caveolins’ functions are compromised, compensatory mechanisms could operate to promote cellular regeneration and survival. Fulizio et al. [78] proposed that regenerating fibers in caveolinopathies muscle have strong overproduction of Cav3. This has two important consequences. The first is that Cav3 overproduction increases the rate of exocytosis, so the total amount of Cav3 at the cell surface is increased and could even be close to normal. The second is that mutations in Cav3 trigger production of the protein with aberrant conformation, which affects its oligomerization and association with lipid rafts. This triggers accumulation of mutant Cav3 at the Golgi apparatus. Accumulation of protein aggregates in the Golgi leads to its increased proliferation and increased rate of exocytosis and repair of the plasma membrane [78,79]. Such accumulation of mutant proteins in the Golgi is an unusual phenomenon as most proteins, when misfolded, accumulate in the endoplasmic reticulum through interaction with chaperones [80].

## 7. Caveolin-3 and T-Tubule Systems

T-tubules (or transverse tubules) are invaginations of the plasma membrane into the center of cardiac and skeletal muscle fibers [81]. T-tubules are abated by two terminal cisternae of the ER and they collectively form a so-called triad [81]. The T-tubules are enriched in the ion channels and transporters and facilitate the transmission of the action potential from the exterior to the interior of the cell. The direct association of T-tubules with the ER through the interactions of the dihydropyridine receptors (DHPRs) on the T-tubules with the ryanodine receptors (RYRs) on the ER membrane allows for rapid and efficient calcium release in response to action potential [81]. Thus, formation and proper functioning of the T-tubules is crucial for fast muscle contraction and synchronization of the contractions in the heart. Caveolin-3 is enriched in T-tubules in skeletal muscles and cardiac myocytes and T-tubule formation depends on functional Cav3 proteins [82]. This is supported by the data from Cav3 null mice which show a disorganized T-tubule system in skeletal muscle and the heart [83,84,85]. Caveolin-3 has been reported to interact with RYR, DHPR, and other T-tubule proteins [81].

Interestingly, T-tubules have approximately four times more cholesterol than the rest of the plasma membrane. Since Cav3 binds cholesterol, they likely actively contribute the formation of the unique biophysical properties of these specialized membrane compartments. The exact mechanisms as to how the T-tubules form is not clear, but this process, apart from Cav3, also involves a BAR domain protein Amfiphysine (Bin1) and actin organizing protein N-WASP [86,87].

## 8. Role of Caveolin-3 in Energy Metabolism

Caveolin-3 can stimulate the phosphorylation of insulin receptor substrate 1 (IRS1), a downstream regulator of an insulin receptor required for glucose uptake in muscles [88]. These receptors have been localized to caveolae at the plasma membrane [89]. The Caveolin-3 also interacts with phosphofructokinase-M (PFK-M), a key enzyme in the control of glycolysis in a glucose dependent manner, and may play a role in the regulation of energy metabolism in skeletal muscle fibers [90]. Shang et al. [91] have shown that in skeletal muscle cell line C2C12, Cav3 protein can activate Akt signaling and increase glucose transporter type 4 (GLUT4) localization at the cell surface, leading to increased glucose uptake. This in turn promotes myocytes’ growth and proliferation. These observations suggest that Cav3 plays an important physiological role in skeletal muscle cell metabolism.

## 9. Caveolin-3 Regulates Lipid Metabolism

Caveolae are abundant in the adipocytes which are cells that store many lipids. It has been shown that lack of caveolae in caveolin-1 knockout mice leads to reduced size of adipocytes and lipodystrophies along with other physiological abnormalities [92]. An ectopically expressed dominant negative mutant of Caveolin1 (CAV^DGV^) promotes complex intracellular lipid imbalance. In particular, expression of CAV^DGV^ increases the level of free cholesterol in late endosomes but depletes cholesterol in the Golgi complex and the plasma membrane [93]. Similarly, human patients with malfunctioning Cav1 mutations develop problems with lipid metabolism and become insulin resistant [94]. It has been reported that Cav1 mediates the transfer of newly synthesized cholesterol from the ER to the plasma membrane and mutations in Cav1 lead to increased sterol levels in the ER [95,96]. Accumulation of cholesterol in the ER would then cause increased acyl-CoA: cholesterol acyltransferase (ACAT) activity that catalyses the esterification of cholesterol and reduced free cholesterol. Collectively, caveolins, which bind cholesterol, serve as a sensor for the cholesterol in the membrane, regulate their intracellular transport, and influence cholesterol metabolism. Cholesterol is an important component of cellular membranes, which regulates many signaling processes. The misbalance of cholesterol levels in the body may cause various diseases, i.e., atherosclerosis [97], stroke [98], coronary heart disease [99], heart attacks [99], and hemorrhagic stroke [100,101].

CD36 is the most important translocase for long-chain unesterified fatty acids, which are the energy source in skeletal muscles [102]. Vistisen et al. [103] have shown that CD36 colocalizes with the Cav3 in sarcolemma, suggesting that caveolae may regulate cellular fatty acid uptake by CD36. Interestingly, Cav3+/− mice have lower CD36 levels in the hearts, which likely leads to decreased lipid uptake in this organ [104]. Moreover, a case study by Ibarretxe et al. [105] showed that Cav3 deficiency leads to severe hypercholesterolemia and also destabilized low-density lipoprotein receptors, pointing to the important role of Cav3 in the regulation of lipid metabolism.

## 10. Function of Caveolin-3 in Mitochondrial Homeostasis

Mitochondrial dysfunction is a hallmark of the pathophysiology of highly metabolic organs like the heart, brain, muscles, or liver. In these organs, mitochondria play important roles in adaptation to cellular stress [106]. Several studies have shown that mitochondria and caveolae can be closely apposed, facilitating their physical and functional interactions [107,108]. Cardiac ischemia, for instance, can induce translocation of Cav1 proteins from caveolae to mitochondria [109,110]. Caveolin-1 has been also enriched in the ER membrane at the sites of contact with mitochondria [111], which appears to be important for Ca^2+^ and lipid homeostasis [112]. The Ca^2+^ released from the ER stimulates pyruvate dehydrogenase in mitochondria and the F0/F1-ATPase required for ATP synthesis, Krebs cycle, and pyruvate decarboxylation [113,114]. Shah et al. [115] have revealed that Cav3 deficiency in cardiac muscle cells associated with the Cav3P104L mutation invokes major disturbances in mitochondrial respiration and energy status that may contribute to the pathology of RMD-2. The targeting of Cav3 to mitochondria improves the function of mitochondria and thus induces stress adaptation in the heart.

## 11. Caveolin-3 Regulates Potassium Channels and Adrenergic Receptors

Caveolin-3 has been reported to regulate TASK1, the two-pore domain potassium channel expressed at the plasma membrane of neurons, cardiac myocytes, and skeletal muscle fibers [116,117,118]. TASK1 channels are in permanently opened confirmation, allowing for K^+^ outward currents at the membrane. This is important for the stabilization of cells below the firing threshold. TASK1 channels play a crucial role in maintaining the resting membrane potential and their malfunction leads to depolarization of the membrane, hyperexcitation of the cell, and subsequent cell death due to excitotoxicity [119,120]. Caveolin-3 inhibits the function of TASK1 and the expression of Cav3 dominant negative mutant (P104L) abolishes TASK1 inhibition.

The beta-adrenergic receptors (βAR) are the G-protein coupled receptors that, in the heart and skeletal muscles, produce cAMP in response to catecholamines, especially adrenalin and noradrenalin, that regulate the pace of heart beats [121]. There are two types of βAR, i.e., β1AR and β2AR [122,123,124]. It has been reported that β2AR is exclusively localized to caveolae [125,126]. Wright et al. [127] have shown that Cav3 plays a crucial role in the function of β2AR in the cardiomyocytes. They have reported that over expressed Cav3 leads to two-fold lower production of cAMP and its retention in caveolae. Conversely, the expression of dominant negative mutant of Cav3, which lowers the number of caveolae, leads to an increase in the cAMP levels.

## 12. Role of Caveolin-3 in Heart Diseases

Caveolae are specialized lipid rafts at the surface of the cardiac and sarcolemma membrane. Therefore, they could play an important role in the overall function of the heart. Moreover, mutations in Cav3 have been detected in various cardiovascular diseases such as arrhythmias, cardiac hypertrophy, and injury after myocardial ischemia/reperfusion. A broad review on this topic has been published elsewhere [128,129,130] and here we summarize only the most important observations.

### 12.1. Arrhythmias

In the cardiac cells, caveolae contain many ion-gated channels such as the primary voltage-gated sodium channel (Nav1.5) [131]. It has been reported that mutations in Cav3 lead to dysfunction of Nav1.5 ion channels observed in inherited arrhythmias such as long QT Syndrome 9 (LQT9) [132]. LQT9 is associated with mutations Thr78Met, Ala85Thr, Phe97Cys, Ser141Arg that affect Cav3 interactions with the Nav1.5 ion channel, resulting in increased late sodium current. These mutations lead to defects in the heart without causing skeletal muscle defects [133]. Moreover, various mutations in Cav3, such as Val14Leu, Thr78Met and Leu79Arg, lead to sudden infant death syndrome, in which a five-fold increase in the late sodium current was observed [134].

### 12.2. Cardiac Hypertrophy

Cardiac hypertrophy is caused by the increase in the size of cardiomyocyte, which ultimately leads to progressive heart failure [135,136]. Several mutations in Cav3, such as Thr63Ser, Thr78Met, Ala46Thr, and the deletion of Phe96, are known to cause cardiac hypertrophy in human patients [128,137,138,139,140]. The Cav3 KO mice show signs of cardiac hypertrophy and reduced function of the heart as characterized by ventricular dilation and reduced contractility [141]. This is further accompanied by cellular infiltration and progressive fibrosis of the heart.

### 12.3. Myocardial Injury after Ischemia/Reperfusion

The overexpression of Cav3 has been shown to protect the heart from ischemia [142]. The heart of Cav3 KO mice show increased contractile dysfunction and cell damage following ischemia [143]. The mechanism of Cav3-mediated protection against ischemia/reperfusion is likely mediated by the activation of the PI3K/Akt/GSK3β pathway [144].

## 13. Interaction of Caveolin-3 with the Dystrophin-Glycoprotein Complexes

Caveolin-3 also interacts with the dystrophin-associated glycoprotein complex (DGC), which is a major laminin receptor on the muscle surface. This multiprotein transmembrane complex plays a pivotal role in the muscle fiber integrity by linking the ECM with cytoskeleton. Additionally, it serves as a scaffold for the recruitment of many signaling molecules. Proteins associated with this complex could be divided into two classes. The core of the complex is composed of α- and β-dystroglycans, dystrophin, utrophin, sarcospan, sarcoglycans, syntrophins, and dystrobrevins [145,146]. Except for the dystrobrevins, the absence of the other core components leads to the disassembly of the entire complex [145,147]. It is, therefore, not surprising that the mutations in most of the core DGC proteins lead to muscular dystrophies. The core of the DGC additionally recruits several peripherally-associated proteins such as signaling molecules nNOS and PI3K, scaffold proteins such as GRB2 and α-Catulin, or cytoskeleton-organizing proteins such as Liprin-α1 and Arhgef5 [148,149,150,151]. For more comprehensive reviews on DGC, see [146,152]. It has been reported that Cav3 is an important interactor of several DGC components including dystrophin, utrophin, and β-dystroglycan.

### 13.1. Caveolin-3 and Dystrophin

Dystrophin is a large (430-kDa), rod-shaped protein, which at the DGC binds β-dystroglycan, syntrophins, and dystrobrevins [153]. This large protein has multiple protein–protein interaction motifs and is known to interact with actin, microtubules, and intermediate filaments as well as signaling molecules like nNOS and certain lipids including cholesterol [154]. Mutation in the dystrophin leads to Duchenne muscular dystrophy (DMD). Dystrophin by its cysteine-rich ZZ domain is attached to the PPXY motif of the β-dystroglycan, anchoring dystrophin’s C terminus to the plasma membrane [155,156]. The WW-like domain of Cav3 also directly interacts with the PPXY motif of the β-dystroglycan [157] competing for the same site with the dystrophin. The association of Cav3 with DG has been proposed to trigger β-dystroglycan degradation (see next paragraphs). Thus, the Cav3 expression level can indirectly impact the expression of the dystroglycan and the stability of the entire DGC.

### 13.2. Caveolin-3 and Utrophin

Utrophin is a dystrophin homolog enriched at the neuromuscular junction in skeletal muscles and it interacts with dystroglycan and dystrobrevins [158,159]. It has been reported that utrophin interacts with Cav1 at the plasma membrane of vascular smooth muscle [160]. This interaction may regulate the activity of eNOS and its overall effect on the cell.

Caveolin-1 interaction with utrophin has been proposed to mediate the internalization of caveolae and the low-density lipoprotein receptor-related protein 1 (LRP1) [161]. LRP1 is a member of the LDL receptor family involved in various biological processes such as lipoprotein metabolism and cell motility, and diseases such as atherosclerosis and cancer. It has been shown that motor protein KIF13B recruits LRP1 to caveolae through utrophin and promotes caveolae internalization through their motor activity and interaction with microtubules [161].

### 13.3. Caveolin-3 and Dystroglycan

There are two dystroglycan subunits—the extracellular α-dystroglycan and the transmembrane β-dystroglycan [162]. They are produced from a single precursor protein as a result of proteolytical cleavage at the Golgi complex. The α-dystroglycan interacts with ECM molecules and the transmembrane β-dystroglycan links the alpha subunit to the cytoplasmic components of the complex. The PPXY motif of β-dystroglycan binds to the WW domain of dystrophin [163] and the WW-like domain of Cav3 in a competitive manner [157]. This interaction appears to be crucial for the regulation of dystroglycan stability and the integrity of the entire DGC. Dystrophin binding to β-dystroglycan stabilizes dystroglycan and the DGC, while Cav3 interaction with β-dystroglycan has been proposed to trigger its degradation and the disassembly of the DGC.

### 13.4. Caveolin-3 and Sarcoglycan

Sarcoglycans are transmembrane glycosylated proteins associated with β-dystroglycan. They have four subunits, i.e., α, β, γ and δ [164]. Mutations in α-, β-, and γ-sarcoglycan have been detected in either autosomal recessive muscular dystrophy, limb-girdle muscular dystrophy, or severe childhood autosomal recessive muscular dystrophy [165]. The α-sarcoglycan complex was found within caveolae membranes in differentiated C2C12 cells, but the direct interaction between sarcoglycans and Cav3 has not been reported [166]. Interestingly, in Cav3 KO heart cells, sarcoglycans are no longer associated with lipid rafts (or, more precisely, detergent-resistant membranes), but their localization to the plasma membrane was not affected. Many DGC components have been reported to associate with caveolae in various cell types. It is, however, less clear what the function of such localization is. Do DGC components re-localize to caveolae before internalization? Are caveolar degradation pathways a route through which DGC is internalized and degraded?

## 14. Caveolin-3 at the Neuromuscular Junction

Muscle fibres and motor neuron axons form specialized synapses called neuromuscular junctions (NMJs) on which the impulse to contract is transmitted. The DGC is enriched at the NMJs, where it performs an important role in stabilising postsynaptic acetylcholine receptors (AChRs) and other components of the postsynaptic machinery on the muscle fibre surface. Caveolin-3 is also concentrated at the postsynaptic specialisation of the NMJ [167] and experimental denervation leads to a decrease in Cav3 levels at the NMJ [167]. Interestingly, Hezel et al. [168] have reported that Cav3 associates with AChRs, and this interaction is strongly enhanced upon treatment with agrin, a glycoprotein secreted by the nerve to promote AChR clustering and assembly of the postsynaptic machinery. Caveolin-3 also binds to the muscle-specific kinase (MuSK), the receptor for agrin on the muscle surface. Remarkably, the lack of Cav3 inhibits the agrin-induced activation of Rac1, which is downstream of MuSK and impairs clustering of AChR [169]. Thus, Cav3 regulates clustering of the postsynaptic machinery, at least in vitro. As Cav3 binds cholesterol and organizes lipid rafts [170,171], one of its functions at the NMJ could be to organise this specialized domain on the surface of muscle fibre. AChRs also bind cholesterol, are enriched in the detergent resistant membranes and the entire postsynaptic assembly was proposed to be a clustered raft [172,173,174,175]. The role of Cav3 at the NMJ may have important implications for pathological processes in various caveolinopathies in which the level of Cav3 decreases in the muscle membrane.

## 15. Muscular Dystrophies Associated with Caveolin-3

Point mutations and dysregulated expression of caveolins have been linked to the pathogenesis of multiple diseases like cancers, cardiovascular disorders, diabetes, and muscular dystrophies [6]. So far, 18 mutations in Cav3 (Figure 1) have been associated with human dystrophies such as limb-girdle muscular dystrophy 1C (LGMD-1C), hereditary rippling muscle disease (RMD), hyperCKemia, and distal myopathy [13], whereas Cav3 overexpression is observed in Duchenne muscular dystrophy. The set of diseases characterized by reduced levels of Cav3 are collectively called caveolinopathies.

### 15.1. Limb-Girdle Muscular Dystrophy

Limb-girdle muscular dystrophy (LGMD) is a clinically and genetically heterogeneous group of myopathies. It has two forms—autosomal dominant form and recessive form. To date, nine autosomal recessive genes for LGMD (LGMD-2A to 2I) and six genes for autosomal dominant form (LGMD-1A to 1F) have been mapped [176]. It has been shown that autosomal dominant limb-girdle muscular dystrophy (LGMD-1C) in humans is caused by mutations in the CAV3 gene. These mutations lead to the loss of Cav3 expression due to degradation of misfolded Cav3 [12]. Recently, a new nomenclature guidance suggested renaming LGMD-1C to Rippling Muscle Disease-2 (RMD2) [177]. RMD2 has been excluded from the list of LGMD as the main clinical features are the rippling muscles and myalgia.

A microdeletion of three amino acids within the caveolin scaffolding domain of Cav3 causes RMD-2 type muscular dystrophy [12]. A point mutation in Cav1 (P132L) leads to the mis-localization and intracellular retention of wild-type Cav1 causing breast cancers in humans in a dominant-negative fashion [14,178]. An analogous P to L mutation in Cav3 (P104L) has been detected in patients with autosomal dominant RMD-2, and this mutation also has a dominant-negative character [12]. Another Cav3 mutation associated with LGMD is a dominant-negative missense mutation Ala45Thr [179]. This mutation leads to the reduced expression of Cav3. Other mutations in Cav3 associated with RMD-2 include Asn32Lys, Gly55Ser, Thr63Pro, Cys71Trp, and Arg125His (Figure 1) [12].

### 15.2. Rippling Muscle Disease

Rippling muscle disease (RMD) is a rare autosomal-dominant disorder of skeletal muscle. It is characterized by signs of increased muscle irritability, such as percussion-induced rapid contraction (PIRC), percussion-induced muscle mounding (PIMM), and/or electrically silent muscle contractions (rippling muscle) [180]. Several mutations in Cav3 protein (Arg26Gln, Ala45Thr, Ala45Val, Asp27Glu, Pro28Leu, Pro28Thr, Val43Glu, Leu86Pro, Ala92Thr, and Pro104Leu) have been associated with RMD [12,18]. Ricci et al. [180] have shown RMD and facioscapulohumeral dystrophy-like phenotypes in a patient carrying a heterozygous Cav3 T78M mutation and a D4Z4 partial deletion. Lo et al. [181] have reported RMD which was associated with reduced Cav3 levels, resulting in partial deficiency, or “mosaicism”, of Cav3 localization in the absence of mutations in the Cav3 coding region.

### 15.3. Asymptomatic hyperCKemia

Asymptomatic hyperCKemia is a type of myopathy showing elevated levels of serum creatine kinase (CK). Creatine kinase is muscle-specific kinase that phosphorylates creatine, which serves as a reserve of high-energy phosphates. If muscle membrane integrity is compromised, then CK leaks out from the muscle fibers and can be detected in blood serum in simple laboratory analyses. Elevated levels of CK in the serum are a hallmark of many muscular disorders, but the asymptomatic hyperCKemia is characterized by lack of apparent muscle weakness. The mutations in Cav3 protein—Arg26Gln [182], Ala45Thr/Val [12] and Pro28Leu [183]—lead to low expression of Cav3 in asymptomatic hyperCKemia [184]. A missense mutation G169A was found in the CAV3 gene in a patient who developed transient hyperCKemia, myalgia, and mild muscular weakness [185]. Bruno et al., have reported a Val44Met mutation in the Cav3 protein of patients with persistent elevation of serum CK levels, myalgia, and hypercholesterolemia [186]. Low expression of Cav3 in this disease leads to decreased caveolae numbers at the plasma membrane resulting in abnormal response to mechanical stress and possible damaged membrane of the muscle fibers. Moreover, cholesterol is an important component of the plasma membrane and it binds with Cav3. Lower levels of cholesterol due to reduced Cav3 in the membrane may also affect the sarcolemma integrity, resulting in the leakage of creatine kinase.

### 15.4. Distal Myopathy

Distal myopathy is a relatively rare subtype of caveolinopathy characterized by weakness of distal muscle groups at the lower or upper limbs. For instance, muscle atrophy restricted to the small muscles of the hands and feet was reported due to a Gly80Ala mutation in Cav3 [187]. The mutation Ala46Thr in Cav3 that turned out to cause a reduced Cav3 expression leads to distal myopathy with hypertrophy of the calf muscles [188].

### 15.5. Duchenne Muscular Dystrophy

Duchenne muscular dystrophy is caused by deficiency in expression of the dystrophin. As mentioned in the previous sections, there is a link between the caveolae, Cav3 expression, and the pathogenesis of DMD. Electron microscopy analysis revealed a three-fold increase in the number of caveolae on the surface of muscle fibers from DMD patients [189]. This turned out to be associated with the elevated expression of Cav3 protein [19]. Similar observations of 2–3-fold elevated levels of Cav3 expression in skeletal muscles were made in mdx mice, an animal model of DMD with a dystrophin deficiency [190]. The elevated level of Cav3 expression is likely a protective response of dystrophin-deficient cells, which have compromised integrity and undergo regeneration [191]. At the same time, elevated levels of Cav3 in the absence of dystrophin could increase Cav3 binding to the β-dystroglycan, enhancing its degradation and disassembly of the DGC [192]. Despite evidence that elevated Cav3 levels could increase the degradation of the DGC, similarly decreased dystroglycan expression has been observed in a patient with Ala46Val dominant negative mutation in Cav3 [179]. The exact functions of Cav3 in the pathology of various muscular dystrophies needs to be further investigated.

## 16. Mice Models of Muscular Dystrophy Associated with Caveolin-3

An increased level of Cav3 protein expression in skeletal muscle fibers of transgenic mice leads to muscular dystrophy, which in many features resembles DMD in humans [192]. These include the presence of numerous hypertrophic fibers undergoing degeneration/regeneration cycles and having characteristic centrally located nuclei associated with this process. Many muscle fibers in these mice undergo necrosis and have increased connective tissues associated with them. The Cav3 transgenic mice also have high levels of serum CK and decreased expression of dystrophin and degradation of the DGC. Mice of the DMD model (mdx mice) do not show proliferation of connective tissues, nor do they show significant necrosis in the muscles [193]. This suggests that Cav3 overexpressing transgenic mice may be a better model for the Duchenne-like dystrophy than mdx mice. Transgenic mice expressing the Pro104Leu mutant of Cav3, which acts in a dominant negative manner, have severe downregulation of the endogenous Cav3 protein and exhibit a myopathic phenotype resembling RMD-2 in humans [17]. A very similar phenotype has been observed in CAV3 knock out mice, which have no caveolae at the sarcolemma and exhibit a mild muscular dystrophy resembling RMD-2 in humans [84,194]. These mice can thus be used as a model for RMD-2 disease in humans.

## 17. Future Perspectives

It has been more than 70 years since caveolae were discovered by G.E. Palade [195]. In recent decades, our understanding of Cav3 functions in skeletal muscles has been widened. Functions of caveolins in endocytosis and formation of a platform for signaling at the cell membrane is still in debate [196]. However, the role of caveolae in muscle and endothelial cells as a protector against the mechanical stress is very well established. These cell types are very much subject to external mechanical stress and the high number of caveolae in these cells help in their function. However, the precise molecular interactions that trigger membrane invaginations and formation of caveolae are not fully understood. It is still not clear how the cells sense and regulate the invagination of the plasma membrane at a specific place. Similarly, the mechanisms that regulate the transition of proteins to and from caveolae require more investigation. Although caveolae are often shaped like little pits at the membrane, they sometimes acquire the form of elongated tubes. What the function of such tubes is and how the shape of these structures is achieved is not clear. Many proteins are known to associate with Caveolins, and some of them have been observed in caveolae. It is not clear, however, to what extent the interactions occur only in caveolae vs. other parts of the membrane and involve binding to monomers or small oligomers. In the past ten years, it has become clear that caveolae and Cav3 regulate muscle functioning, and their altered expression leads to muscular dystrophies. However, further studies are necessary to elucidate the molecular interactions associated with Cav3, which lead to muscular dystrophies. The Cav3-deficient mouse models and/or Cav3 overexpressing transgenic mice will be useful models to study muscular dystrophies—in particular, how a single mutation in Cav3 leads to many forms of muscular dystrophies. For instance, the mutation R26Q has been associated with distal myopathy, RMD and hyperCKemia and the A45T/V mutation leads to RMD-2, RMD and hyperCKemia. Another interesting aspect to study is the role of Cav3 high expression in muscles of mdx mice and DMD patients. The exact function of caveolae in DMD is still a debatable issue. The role of Cav3 in the maintenance of sarcolemma integrity and regulation of dystroglycan degradation in DMD needs further research. Recent reports suggest the involvement of utrophin, an important DGC component in caveolar internalization and intracellular transport. While it is certain that caveolae protect cells from mechanical damage, it is still not clear how the cells sense the tensions on the membrane and induce caveolae disassembly to release the reservoir of the membrane. These findings may unravel novel mechanisms implicated in the pathology of muscles and could serve a role in designing new therapeutic strategies for muscular dystrophies.

## Figures and Tables

**Figure 1 ijms-21-08736-f001:**
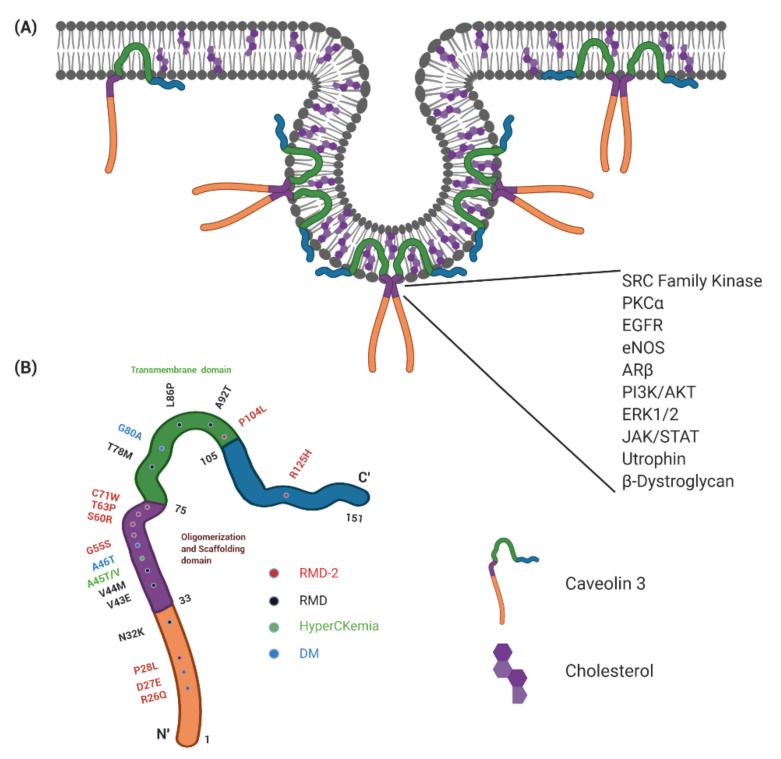
Caveolin-3 domain organization and its localization to caveolae. (**A**) A diagrammatic representation of caveolae with associated Caveolin-3 protein. Caveolae are rich in cholesterol and provide platforms for various signaling molecules like GPCRs, Src kinase, etc. that interact with Caveolins’ scaffold domain (shown on the right). The plasma membrane also contains Caveolin-3, which is not associated with caveolae. (**B**) Mutations in Caveolin-3 causing caveolinopathies. The figure was created with BioRender.com.

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
