# Peer review of "A Role for Caveolin-3 in the Pathogenesis of Muscular Dystrophies"

_ijms, 2020, doi:10.3390/ijms21228736_

Round 1

Reviewer 1 Report

The article is clear and well structured with appropriate references.

However authors should reconsider paragraph 9 " Caveolin-3 regulates lipid metabolism " in which the role of Cav-1 is described without mentioning Cav-3. In particular, the metabolic interaction between the membrane defect associated with Cav-3 deficiency and the alterations of lipid metabolism should be evaluated.

Reviewer 2 Report

The review title “A Role for Caveolin-3 in the Pathogenesis of Muscular Dystrophies” Here In this review, the Authors summarize the significant functions of Caveolin-3 in skeletal muscles and discuss its involvement in the pathology of muscular dystrophies. This review provides our understanding of the structure and function of Caveolin-3 in the pathology of muscles and it´s role in muscular dystrophies. It was more or less convincing. I prefer this article is suitable for publication but after some minor corrections.

  • Could please the authors elaborate or add more information on 6. Function of Caveolin-3 in muscle repair and development part.

Author Response

Response to Reviewers’ Comments

Dear Reviewers,

We are very grateful for your thorough reading of our manuscript “A Role for Caveolin-3 in the Pathogenesis of Muscular Dystrophies by Pradhan and Prószyński. We appreciate your suggestions, which we addressed carefully. We believe that our manuscript is significantly improved thanks to your comments. We hope that you will find that our manuscript now meets the necessary criteria for publishing in the International Journal of Molecular Sciences.

Reviewer 2

Could please the authors elaborate or add more information on 6. Function of Caveolin-3 in muscle repair and development part.

We have now added more information about the role of Cav3 in muscle repair and development (line no 160 - 174).

Reviewer 3 Report

This is a rather extensive review on the role of caveolin-3 in various disorders expecially in muscle disorders:the only part not extensively covered for caveolin-3 is its role in muscle regeneration,in fact even in caveolinopathy some regenerating fibers express again caveolin-3,as demonstrated by Fulizio et al.(2004).This expression is apparent in particularly in relationship to golgi apparatus and rthis should be adequately explained and commented.

Recently there has been a revision on LGMD nomenclature by an ENMC consortium published by Straub (2018) in NMD and therefore the term LGMD1C should be substituted by the term caveolinopathy with the corresponding clinical phenotypic expression.The authors can also use both classifications but should then explain that this entity is no more part of dominant LGMD.

Another caveat and caution in interpretation should be introduced in the statements in introduction and conclusion regarding DMD.It is necessary to further document the role of caveolin-3 in human pathology and its possible to speculateon a positive or negative role for determining myopathological changes or increased fibrosis,since pathology is variable in human dystrophies and often different from from animal models.It is necessary that in introduction the term malfunction should be better explained in Duchenne muscular dystrophy were the main feature is an overexpression (?) but it is unknown for what possible adptive changes this is meaningful in term of  function,many hypertrophic fibers could be either be reactive or loaded with calcium .In fact is possible that increased caveolin expression could be part of pathology,although most of the work has been done in animal models,so a caveat is necessary applyiing such models to human dystrophy.

Authors should also explain the role of caveolin-3 in detrmining Hyperckemia syndrome (see papers by Kiriakides et al,2013,2020,Muscle&Nerve)

Author Response

Response to Reviewers’ Comments

Dear Reviewers,

We are very grateful for your thorough reading of our manuscript “A Role for Caveolin-3 in the Pathogenesis of Muscular Dystrophies by Pradhan and Prószyński. We appreciate your suggestions, which we addressed carefully. We believe that our manuscript is significantly improved thanks to your comments. We hope that you will find that our manuscript now meets the necessary criteria for publishing in the International Journal of Molecular Sciences.

Reviewer 3

  1. a) …the only part not extensively covered for caveolin-3 is its role in muscle regeneration, in fact even in caveolinopathy some regenerating fibers express again caveolin-3, as demonstrated by Fulizio et al.(2004). This expression is apparent in particularly in relationship to golgi apparatus and this should be adequately explained and commented.

We have now added more information about the role of Cav3 in muscle regeneration (line no 171 - 173).

(b) Recently, there has been a revision on LGMD nomenclature by an ENMC consortium published by Straub (2018) in NMD and therefore the term LGMD1C should be substituted by the term caveolinopathy with the corresponding clinical phenotypic expression. The authors can also use both classifications but should then explain that this entity is no more part of dominant LGMD.

The nomenclature has been corrected (line no 351 - 354).

(c) Another caveat and caution in interpretation should be introduced in the statements in introduction and conclusion regarding DMD. It is necessary to further document the role of caveolin-3 in human pathology and its possible to speculate on a positive or negative role for determining myopathological changes or increased fibrosis, since pathology is variable in human dystrophies and often different from animal models. It is necessary that in introduction the term malfunction should be better explained in Duchenne muscular dystrophy were the main feature is an overexpression (?) but it is unknown for what possible adaptive changes this is meaningful in term of function, many hypertrophic fibers could be either be reactive or loaded with calcium. In fact is possible that increased caveolin expression could be part of pathology, although most of the work has been done in animal models, so a caveat is necessary applying such models to human dystrophy.

We have now wrote this more clearly in the text (line no 41 - 51).

(D) Authors should also explain the role of caveolin-3 in detrmining Hyperckemia syndrome (see papers by Kiriakides et al,2013,2020,Muscle&Nerve)

We have now explained this in the text (line no 389 - 392).

Once again, let me express my gratitude for your helpful comments and suggestions.

Best regards,

Tomasz J. Proszynski

Round 2

Reviewer 3 Report

This is a rather extensive review on Caveolinopathy a term that covers various type of clinical muscle disorders, ranging from hyperCKemia syndrome to proximal and distal myopathies,rippling muscle disease causing four types of muscle disorders even in the same family .The human genetic disorder is dominant but also acquired  disorders are sometime accompanied by rippling phenomenon an interesting disorder with peculiar ondulating waves of contraction at the surface of skeletal muscles.Caveolin-3 mutants have been described in patients with hypertrophic cardiomyopathies and congenital long QT syndrome with mutations in caveolin-3 gene have been identified(Gazzero,et al 2011,Handbook of Clinical Neurology 2011)It would be important to mention also caveolin role in heart disorders.Several hypertrophic cardiomyopathies were in fact linked to 3p25.Caveolae are specialized lipid rafts at surface of the cardiac and sarcolemma membrane.

Caveolin-3 is restricted to striated cardiac or skeletal muscle, but its role is still incompletely studied and links several subcellular cell compartments like the Golgi apparatus, were it originates the T-Tubule and sarcolemma ,constitute high specific macromolecular complexes perturbed in caveolinopathy cases .Caveolae internalization repairs wounded cell and myofibers.
The reappearance of caveolin-3 in regeneration in fibers undergoing regeneration (as demonstrated by fetal myosin stain) is another interesting phenomenon.

The relevant role of caveolaeis is a debatable issue in Duchenne Muscular Dystrophy were their key role in sarcolemma maintenance still needs a full explanation and is perplexing, however it has to be remembered that historically what brought interest to muscle was the scarcity of caveolae observed in DMD muscle by Bonilla, Fishbeck and Schotland (Am.J.Pathol.1981)with freeze-fracture studies.This could also be mentioned.

Author Response

Please check the attachemnt. 
